# Residential area deprivation and risk of subsequent hospital admission in a British population: the EPIC-Norfolk cohort

Robert Luben [ORCID],[1] Shabina Hayat,[1] Anthony Khawaja,[2] Nicholas Wareham,[3] Paul P Pharoah,[1] Kay-Tee Khaw[1]

[1]Department of Public Health and Primary Care, University of Cambridge School of Clinical Medicine, Cambridge, UK
[2]NIHR Biomedical Research Centre, Moorfields Eye Hospital, London, UK
[3]MRC Epidemiology Unit, University of Cambridge School of Clinical Medicine, Cambridge, Cambridgeshire, UK

**Correspondence to**
Dr Robert Luben;
robert.luben@phpc.cam.ac.uk

## ABSTRACT

**Objectives** To investigate whether residential area deprivation index predicts subsequent admissions to hospital and time spent in hospital independently of individual social class and lifestyle factors.

**Design** Prospective population-based study.

**Setting** The European Prospective Investigation into Cancer in Norfolk (EPIC-Norfolk) study.

**Participants** 11 214 men and 13 763 women in the general population, aged 40–79 years at recruitment (1993–1997), alive in 1999.

**Main outcome measure** Total admissions to hospital and time spent in hospital during a 19-year time period (1999–2018).

**Results** Compared to those with residential Townsend Area Deprivation Index lower than the average for England and Wales, those with a higher than average deprivation index had a higher likelihood of spending >20 days in hospital multivariable adjusted OR 1.18 (95% CI 1.07 to 1.29) and having 7 or more admissions OR 1.11 (95% CI 1.02 to 1.22) after adjustment for age, sex, smoking status, education, social class and body mass index. Occupational social class and educational attainment modified the association between area deprivation and hospitalisation; those with manual social class and lower education level were at greater risk of hospitalisation when living in an area with higher deprivation index (p-interaction=0.025 and 0.020, respectively), while the risk for non-manual and more highly educated participants did not vary greatly by area of residence.

**Conclusion** Residential area deprivation predicts future hospitalisations, time spent in hospital and number of admissions, independently of individual social class and education level and other behavioural factors. There are significant interactions such that residential area deprivation has greater impact in those with low education level or manual social class. Conversely, higher education level and social class mitigated the association of area deprivation with hospital usage.

## Strengths and limitations of this study

► This study is able to examine hospital activity using a prospective cohort design in a population of community-dwelling participants with clearly defined population denominators.
► It uses a large cohort of middle-aged and older men and women with 19 years of follow-up time and detailed measurements of demographic and behavioural indicators.
► Both area-based census measures and individual social class and education level from questionnaires are used.
► Differential misclassification in hospital use may be explained by early death rates.
► Socioeconomic determinants of hospitalisation were examined in the context of UK National Health Service hospitals, which are free at the point of use and so not directly influenced by income.

## INTRODUCTION

The considerable differences in mortality by social class are well documented[1–4] with those in higher social classes having a typical life expectancy several years longer than those with the lowest. Similarly, life expectancy and health expectancy varies between UK cities and regions with large variations in expected years of life in good health.[5 6] Despite increasing overall life expectancy, inequality remains with lower life and health expectancy observed more often in disadvantaged groups. While lifestyle factors may account for some part of this, the reported differences in death rates cannot be explained by individual behaviour alone.[7 8] Material deprivation was defined by Townsend as 'a state of observable and demonstrable disadvantage relative to the local community or wider society … to which an individual, family or group belongs'. Deprivation indices use factors such as unemployment, the standard of housing, overcrowding and rates of car ownership which together can assess the level of deprivation within a neighbourhood.[9]

Hospitalisation can be measured using the frequency of admission or the length of stay. When measured over a period of time, the

outcome represents burden of resources that might be attributable to a population. Inequality in healthcare utilisation favouring patients who are better off is apparent in half of the Organisation for Economic Co-operation and Development countries.[10–12] The UK National Health Service is free at the point of use and consequently should provide equitable healthcare not constrained by ability to pay.

Socioeconomic determinants of hospitalisation have been examined using individual level exposures such as occupational social class, income and education and at area level using various deprivation indices but few studies have both individual and area-based measures. Individual occupational social class, income and level of education have all been reported to be associated with chronic disease risk.[13 14] We previously reported that a range of simple demographic and behavioural indicators are related to the future probability of cumulative hospital admissions and bed days.[15] Increasing age and male sex and the modifiable factors current cigarette smoking, body mass index (BMI) $>30 \, kg/m^2$, manual social class and low education level were all associated with higher future hospital usage over a 10-year period. Area-based deprivation measures, available routinely in the UK using postal code linkage, have also reported associations with hospital usage.[16–20] However, the participants in such studies are often limited to those attending hospital and so a suitable population denominator is lacking. Studies reporting health associations for both individual and area measures are less common[21–23] and we are unaware of any studies examining the independent association of residential area deprivation on subsequent hospital usage.

In this paper, we examine residential area deprivation using the Townsend Area Deprivation Index with subsequent hospital usage over a 19-year period. We explore the independent contribution of residential area deprivation in men and women participants of the European Prospective Investigation into Cancer in Norfolk (EPIC-Norfolk) study and its association with future hospitalisation after allowing for the individual level factors previously shown to be associated. We also examine possible interactions between area and individual deprivation measures. Our aim is to determine whether factors such as material living conditions, poor quality housing and poor infrastructure are associated with subsequent hospitalisation in a setting where access to healthcare is unconstrained by ability to pay.

## METHODS
We used data collected as part of EPIC-Norfolk, a general population cohort.

### Study design
EPIC-Norfolk is a cohort of men and women living in Norfolk. Recruitment took place between 1993 and 1997 at 35 general practices with invitations sent to all those registered with the practices aged 40–79 years. The

design and recruitment of the study has been previously described in detail.[24 25] Briefly, a total of 77 630 invitations were sent to adults registered at participating GP practices; 30 445 (40%) consented to participate in the study of whom 25 639 men and women completed a lifestyle questionnaire and attended a health examination. Residential postcode, recorded at the end of recruitment, was used to link to the UK 1991 national census data.[26] Between 1999 and 2018, the cohort was linked to databases maintained by the East Norfolk Primary Care Trust (PCT) and later to national databases held by NHS Digital.[27] Hospital Episode Statistics (HES) records which included admission and discharge dates were used to calculate time in hospital and number of admissions. Contiguous admissions were merged and counted as a single admission. Details of linkage of the EPIC-Norfolk cohort participants to hospital records have been previously reported.[15]

### Residential area deprivation score for participants
The Townsend Area Deprivation Index is an area deprivation measurement calculated using four components: the percentage unemployed of economically active residents aged over 16 years, the percentage of households with no car, the percentage of households not owner occupied and the percentage of households with more than one person per room. These are respectively: a measure of lack of material resources and insecurity, a proxy for current income, a proxy for current wealth and a measure of material living conditions. The index used in this study was constructed using data collected at the 1991 UK census, which takes place every 10 years. Each Townsend component was calculated at Enumeration District (ED), a small area containing an average 175 households used by the census administrators both as output areas and for data collection. Townsend components were then standardised as z-scores at ED level for England and Wales. Study participants were linked to an ED using their home postcode at the end of recruitment in the year 2000. The link was then used to establish a residential Townsend Area Deprivation Index for each individual.

### Covariables
Participants' height and weight were measured in light clothing without shoes by trained nurses in a clinic setting as part of a health examination between 1993 and 1997. Height was measured to the nearest 0.1 cm using a stadiometer (Chasemores, UK) and weight to the nearest 0.1 kg BMI was calculated using measured weight in kilograms divided by measured height² in square metres.

Participants completed a lifestyle questionnaire which included questions about their and their partner's current and past employment. Occupational social class was defined according to the Registrar General's classification[28] and dichotomised into non-manual and manual social classes. Professional, managerial and technical and non-manual skilled occupations (codes I, II and IIIa, respectively) were classed as non-manual while manual

skilled, partly skilled and unskilled (codes IIIb, IV and V, respectively) were classed as manual. Social class for men used (in order of priority) their own current employment, own past employment, partner's current employment or partner's past employment according to whether a social class classification could be defined for a given occupation type. Similarly, social class for women used (in order of priority) their partner's current employment, partner's past employment, own current employment, own past employment. The use of partner's social class for women in the EPIC-Norfolk cohort born between 1918 and 1948 has been previously discussed.[29]

The question "Do you have any of the following qualifications" together with a list of common UK qualifications was used to establish educational attainment. Participants were categorised according to the highest qualification they attained: those with no formal qualifications, those with formal qualifications usually associated with completing school aged between 16 and 18 years and those with degree level qualifications.

Smoking status was derived from two questions: "Have you ever smoked as much as one cigarette a day for as long as a year" and "Do you smoke cigarettes now". The responses to both questions were 'yes' or 'no' and participants were asked to leave the second question blank if they answered 'no' to the first.

Travel time and travel distance between participants home postcode and the Norfolk and Norwich University Hospitals NHS Foundation Trust was calculated using the Open Source Routing Machine,[30] which calculates the shortest path between two points over the road network. Postcode of home residence was used to establish if a participant had moved house over the follow-up period. It was available at two points in time: in the year 2000 and the year 2014. Participants whose postcode or house location changed during the period were classified as having moving house but were not excluded from the analyses. Urban and rural categories were established using the 1991 census.

### Ascertainment of hospital usage and mortality through record linkage

Details of linkage of the EPIC-Norfolk cohort participants to hospital records have been previously reported.[15] Briefly, linkage using unique NHS numbers was performed between 1999 and 2018 to databases maintained by the East Norfolk Primary Healthcare Trust and to national databases held by NHS Digital.[27] All hospital activity for EPIC-Norfolk participants was captured wherever they were treated in England and Wales. HES records which included admission and discharge dates were used to calculate time in hospital and number of admissions. Contiguous admissions were merged and counted as a single admission.

### Statistical analysis

For the current analyses, we excluded the 625 men and women from the baseline cohort who died before 1999.

A further 37 who did not have a valid UK postcode were excluded leaving 24 977 participants. Dichotomous variables were created for the three socioeconomic status variables. Occupational social class was categorised into non-manual and manual: social classes I, II and III non-manual were classified as 'non-manual', while social classes III manual, IV and V were classified as 'manual'. Educational level was categorised into 'higher level' (which includes those with qualifications at secondary level or above) and 'lower level' (those with no qualification). Townsend Area Deprivation Index was divided into quintiles. Lower Townsend scores correspond to lower levels of deprivation. Quintiles 1–4 are all below zero and hence below (less deprived than) the national average for England and Wales. Quintile 5 (−0.64, 6.99] corresponds to Townsend scores close to or above the national average (more deprived). Overall Townsend score and components were also dichotomised with scores below zero defined as 'less deprived' and scores above 0 as 'more deprived'. Hospital admissions were categorised into five groups: 0, 1, 2–3, 4–6 and ≥7 while time in hospital was divided into categories: none, day case, 2–5 days, 6–20 days and >20 days. The cut-points were chosen to be consistent with earlier work.[15] Since time in hospital was skewed with some people remaining in hospital for extended periods, length of stay longer than 365 days was truncated for graphical presentation. A dichotomous urban/rural variable was defined with 'urban' and 'urban sparse' as urban and 'town', 'village' or 'hamlet' as rural. Three dichotomous outcome categories were calculated: any hospital admissions (vs no admissions), 7 or more admissions (vs fewer than 7) using total admissions and >20 hospital days (vs 20 or fewer) using total bed days (overnight stays) and day cases. Multivariable logistic regression was used for all models. All analyses were performed using the R statistical language (R Foundation for Statistical Computing, Vienna, Austria, V.3.5.3 with packages knitr, Gmisc, ggplot2, tidyverse, intubate).

### RESULTS

Table 1 shows descriptive characteristics by quintiles of residential Townsend Area Deprivation Index for 11 214 men and 13 763 women. The majority (n=20 996) of study participants had deprivation index below zero while n=3 981, approximately corresponding to those in quintile 5, had levels above the national average. Participants in quintile 5 were much more likely to live in an urban setting (70.2%) while those in quintiles 2, 3 and 4 were more likely to live in a rural location. Travel distance was lowest for participants in quintile 1 and 5, perhaps due to a higher proportion living in cities and travel times followed a similar pattern. Participants in quintile 5 were the most likely to move house (26.1% between 2000 and 2014). Hospital admissions and time in hospital are shown for both the full cohort and restricted to those who attended hospital; 10.5% of study participants had no admissions over the 19 years from 1999 to 2018.

**Table 1** Descriptive characteristics by quintiles of Townsend Area Deprivation Index

| | Total | Quintile 1 (−6.74, −3.81) | Quintile 2 (−3.81, −2.94] | Quintile 3 (−2.94, −2.09] | Quintile 4 (−2.09, −0.64] | Quintile 5 (−0.64, 6.99] | P value |
|---|---|---|---|---|---|---|---|
| **Sex (n (%))** | | | | | | | |
| Men | 11 214 (44.9) | 2271 (45.2) | 2262 (45.4) | 2280 (45.2) | 2226 (45.0) | 2175 (43.7) | 0.41 |
| Women | 13 763 (55.1) | 2752 (54.8) | 2723 (54.6) | 2760 (54.8) | 2722 (55.0) | 2806 (56.3) | |
| **Age, years** | | | | | | | |
| Mean±SD | 59.0±9.3 | 58.8±9.0 | 59.0±9.2 | 58.8±9.2 | 59.2±9.4 | 59.4±9.5 | 0.002 |
| **Body mass index, kg/m²** | | | | | | | |
| Mean±SD | 26.4±3.9 | 26.1±3.8 | 26.3±3.8 | 26.4±3.9 | 26.5±4.0 | 26.5±4.1 | <0.001 |
| **Cigarette smoking (n (%))** | | | | | | | |
| Current | 2895 (11.7) | 457 (9.2) | 501 (10.1) | 569 (11.4) | 575 (11.7) | 793 (16.1) | <0.001 |
| Former | 10 411 (42.0) | 2033 (40.7) | 2083 (42.1) | 2044 (41.0) | 2132 (43.4) | 2119 (43.1) | |
| Never | 11 453 (46.3) | 2502 (50.1) | 2361 (47.7) | 2378 (47.6) | 2203 (44.9) | 2009 (40.8) | |
| **Social class dichotomised (n (%))** | | | | | | | |
| Non-manual | 14 691 (60.1) | 3336 (67.4) | 3170 (64.8) | 2950 (59.8) | 2840 (58.9) | 2395 (49.5) | <0.001 |
| Manual | 9741 (39.9) | 1610 (32.6) | 1722 (35.2) | 1985 (40.2) | 1982 (41.1) | 2442 (50.5) | |
| **Level of education (n (%))** | | | | | | | |
| Higher level | 15 841 (63.5) | 3439 (68.5) | 3373 (67.7) | 3218 (63.9) | 3084 (62.4) | 2727 (54.8) | <0.001 |
| Lower level | 9118 (36.5) | 1584 (31.5) | 1611 (32.3) | 1819 (36.1) | 1858 (37.6) | 2246 (45.2) | |
| **Travel distance to hospital, km** | | | | | | | |
| Mean±SD | 20.4±13.1 | 16.5±11.3 | 20.6±12.1 | 22.0±12.2 | 25.2±13.2 | 17.5±14.5 | <0.001 |
| **Travel time to hospital, min** | | | | | | | |
| Mean±SD | 20.8±10.3 | 18.0±8.9 | 20.8±9.5 | 21.9±9.4 | 24.4±10.6 | 19.0±11.6 | <0.001 |
| **Urban or rural location (n (%))** | | | | | | | |
| Urban | 11 214 (44.9) | 2500 (49.8) | 1832 (36.8) | 1810 (35.9) | 1575 (31.8) | 3497 (70.2) | <0.001 |
| Rural | 13 763 (55.1) | 2523 (50.2) | 3153 (63.2) | 3230 (64.1) | 3373 (68.2) | 1484 (29.8) | |
| **Moved house between 2000 and 2014 (n (%))** | | | | | | | |
| Moved house | 5355 (22.2) | 963 (19.8) | 972 (20.4) | 1091 (22.4) | 1060 (22.4) | 1269 (26.1) | <0.001 |
| Did not move house | 18 728 (77.8) | 3903 (80.2) | 3799 (79.6) | 3774 (77.6) | 3662 (77.6) | 3590 (73.9) | |
| **Deaths prior to March 2018 (n (%))** | | | | | | | |
| Dead | 8727 (35.0) | 1630 (32.5) | 1704 (34.3) | 1703 (33.9) | 1781 (36.1) | 1909 (38.4) | <0.001 |
| Alive | 16 198 (65.0) | 3386 (67.5) | 3270 (65.7) | 3327 (66.1) | 3155 (63.9) | 3060 (61.6) | |
| **Hospital activity 1999–2018** | | | | | | | |
| No admissions | 2628 (10.5) | 543 (10.8) | 528 (10.6) | 539 (10.7) | 559 (11.3) | 459 (9.2) | 0.011 |
| One or more admissions | 22 316 (89.5) | 4476 (89.2) | 4449 (89.4) | 4494 (89.3) | 4383 (88.7) | 4514 (90.8) | |
| 7 or more admissions | 16 497 (66.1) | 3417 (68.1) | 3295 (66.2) | 3332 (66.2) | 3291 (66.6) | 3162 (63.6) | <0.001 |
| >20 hospital days | 15 144 (60.7) | 3185 (63.5) | 3054 (61.4) | 3097 (61.5) | 2959 (59.9) | 2849 (57.3) | <0.001 |
| **Time spent in hospital 1999–2018, days** | | | | | | | |
| Full cohort 1999–2018, mean±SD | 32.8±63.0 | 30.5±54.9 | 33.0±69.0 | 31.2±57.2 | 32.5±62.9 | 37.0±69.6 | <0.001 |
| Hospital attenders 1999–2018, mean±SD | 36.7±65.6 | 34.2±57.0 | 36.9±72.0 | 34.9±59.5 | 36.6±65.7 | 40.8±72.0 | <0.001 |
| **Number of inpatient admissions 1999–2018** | | | | | | | |
| Full cohort 1999–2018, mean±SD | 7.5±26.0 | 7.3±22.8 | 8.2±36.1 | 7.6±30.8 | 6.7±11.7 | 7.6±21.4 | 0.073 |
| Hospital attenders 1999–2018, mean±SD | 8.4±27.3 | 8.2±24.0 | 9.2±38.1 | 8.6±32.5 | 7.6±12.1 | 8.4±22.3 | 0.095 |

Round brackets in intervals denote strict inequalities (< or >); square brackets denote non-strict inequalities (≤ or ≥).

**Table 2** Multivariable logistic regression of risk factors by quintiles of Townsend Area Deprivation Index for any hospital admissions, ≥7 hospital admissions and >20 days of hospital stay from 1999 to 2018 in 24 977 men and women and in a subset of 16 198 men and women alive in March 2018

| | Quintile 1 (−6.74, −3.81) | Quintile 2 (−3.81, −2.94] | Quintile 3 (−2.94, −2.09] | Quintile 4 (−2.09, −0.64] | Quintile 5 (−0.64, 6.99] | P (trend) |
|---|---|---|---|---|---|---|
| Outcome of any hospital admissions | | | | | | |
| Model 1 | 1.00 | 1.02 (0.89–1.16) | 1.01 (0.89–1.15) | 0.93 (0.82–1.05) | 1.17 (1.02–1.34) | 0.175 |
| Model 1* | 1.00 | 1.05 (0.91–1.21) | 1.03 (0.89–1.18) | 0.95 (0.83–1.09) | 1.25 (1.08–1.45) | 0.056 |
| Model 2 | 1.00 | 1.00 (0.88–1.14) | 1.00 (0.87–1.14) | 0.90 (0.79–1.03) | 1.09 (0.95–1.26) | 0.731 |
| Model 2* | 1.00 | 1.04 (0.90–1.20) | 1.01 (0.88–1.16) | 0.91 (0.79–1.05) | 1.18 (1.01–1.37) | 0.341 |
| Outcome of 7 or more hospital admissions | | | | | | |
| Model 1 | 1.00 | 1.08 (0.99–1.18) | 1.09 (1.00–1.18) | 1.05 (0.97–1.15) | 1.20 (1.10–1.31) | <0.001 |
| Model 1* | 1.00 | 1.06 (0.95–1.19) | 1.15 (1.03–1.29) | 1.15 (1.03–1.29) | 1.39 (1.25–1.56) | <0.001 |
| Model 2 | 1.00 | 1.07 (0.98–1.16) | 1.05 (0.96–1.15) | 1.01 (0.92–1.10) | 1.11 (1.02–1.22) | 0.107 |
| Model 2* | 1.00 | 1.05 (0.93–1.17) | 1.11 (0.99–1.24) | 1.09 (0.97–1.22) | 1.28 (1.14–1.43) | <0.001 |
| Outcome of >20 hospital days | | | | | | |
| Model 1 | 1.00 | 1.09 (1.00–1.19) | 1.10 (1.00–1.20) | 1.14 (1.04–1.24) | 1.27 (1.17–1.39) | <0.001 |
| Model 1* | 1.00 | 1.08 (0.96–1.22) | 1.15 (1.02–1.30) | 1.23 (1.09–1.40) | 1.45 (1.28–1.64) | <0.001 |
| Model 2 | 1.00 | 1.08 (0.99–1.18) | 1.05 (0.96–1.15) | 1.09 (1.00–1.19) | 1.18 (1.07–1.29) | 0.001 |
| Model 2* | 1.00 | 1.06 (0.94–1.21) | 1.09 (0.96–1.23) | 1.16 (1.03–1.32) | 1.34 (1.18–1.51) | <0.001 |

Model 1: adjusted for age and sex. Model 2: adjusted for age, sex, manual social class, low education, current cigarette smoker, body mass index >30 kg/m$^2$.
*Excluding participants who died prior to April 2018.

Table 2 shows the multivariable logistic regression for quintiles of Townsend Area Deprivation Index and three outcomes: any hospital admissions, ≥7 hospital admissions and >20 days of hospital stay between 1999 and 2018. Model 1 is adjusted for age and sex while model 2 is additionally adjusted for manual social class, lower education level, current cigarette smoking and BMI >30 kg/m$^2$. Additionally, each model is repeated in the subset of participants who survived to the end of the follow-up period. Compared to those with Townsend Area Deprivation Index quintiles 1–4 (lower than the average for England and Wales), those with a deprivation index in quintile 5 had a higher risk of spending >20 days in hospital multivariable adjusted OR 1.18 (95% CI 1.07 to 1.29) and for 7 or more hospital admissions OR 1.11 (95% CI 1.02 to 1.22), but there was no association for any hospital admission. The multivariable adjusted p value for trend across quintiles of Townsend Area Deprivation Index was 0.001 for >20 hospital days and 0.107 for 7 or more admissions. Associations in the subset of participants surviving to March 2018 (n=16 198) were higher than those for the full cohort.

Figure 1 shows graphs of length of hospital stay by quintiles of Townsend Area Deprivation Index and demonstrates the disparity between individual socioeconomic factors and hospital stay when area deprivation index is also considered. In the first plot, results are stratified by higher and lower educational attainment. The difference in days between the least deprived (quintile 1) and the

most deprived (quintile 5) is 6 days for those with lower educational attainment and 3 days for those with higher educational attainment. The second plot shows results stratified by manual and non-manual social class. The difference in days between the least deprived and the most deprived is 8 days for those with a manual social class and 3 days for those with a non-manual social class. Significant interactions were observed between social class, level of education and Townsend Area Deprivation Index (p-interaction=0.025 and 0.020, respectively).

Online supplementary table 1 shows the multivariable logistic regression for risk factors for outcomes of any hospital admissions, ≥7 hospital admissions and >20 days of hospital stay between 1999 and 2018. Models are presented for all participants, men and women and each risk factor is adjusted for all others for the nine models. Male sex is only included in the models for all participants. Age, social class, education BMI and the four individual components of Townsend Area Deprivation Index are modelled. As previously reported, age, male sex, lower education level, manual social class, current cigarette smoking and a BMI >30 kg/m$^2$ were all associated with increased hospitalisation. No single component of the Townsend Area Deprivation Index was more strongly associated for all outcomes for both men and women. However, the unemployment component was associated with seven or more hospital admissions. Areas with low car ownership appeared to have a greater association in women than men.

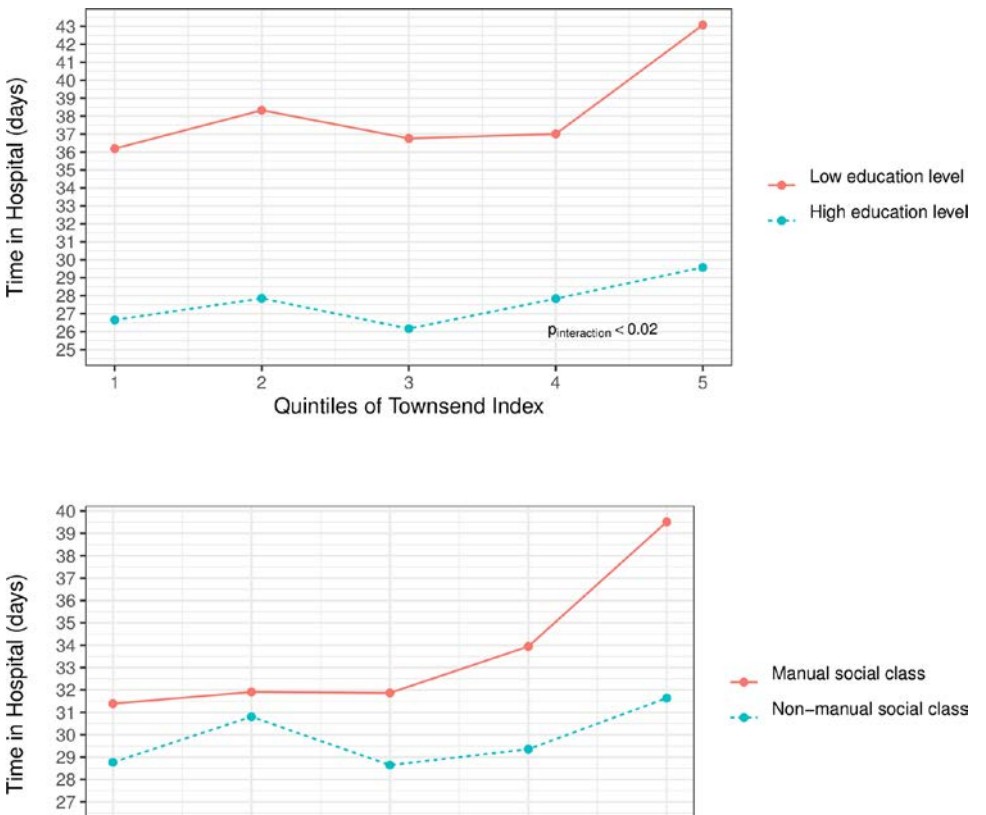

**Figure 1** Hospitalisation by deprivation index. Length of hospital stay over 19 years of follow-up by quintiles of Townsend Area Deprivation Index grouped by categories of education level and categories of social class. Low education level is defined as those having no qualifications and high education as those with at least some qualifications at secondary level or above. Length of stay is truncated to 365 days for those staying longer than 365 days. Interaction tested using multivariable adjusted linear regression with covariables age, sex, education level (higher/lower), body mass index ($\leq$30/>30 kg/m$^2$), smoking status (current/non-current).

Online supplementary table 2 displays logistic regression models for the outcome of >20 hospital days for Townsend Area Deprivation Index in various subgroups. Models are stratified by a dichotomised subgroup: men and women, age above or below 65 years, manual and non-manual social class, lower or higher education level, smoking status, BMI above and below 30 kg/m$^2$, urban or rural home postcode and moved house between the year 2000 and 2014. ORs within all strata were in consistent directions with no interaction by age, smoking status or BMI.

The numbers of individuals with missing values for covariables were: 53 BMI, 218 smoking status, 545 social class, 18 education level.

## DISCUSSION

Residential area deprivation was associated with future hospital usage independently of individual sociodemographic factors, in particular age, sex, social class and education as well as lifestyle factors including smoking and BMI in this cohort of middle-aged and older men and women. Study participants in the highest fifth of the

Townsend Area Deprivation Index—those living in the most deprived areas, at or below the national average, were more likely to spend >20 days in hospital or be admitted to hospital on >7 occasions. There were also significant interactions between residential area deprivation and individual social class and education level. Participants with a manual social class living in an area with higher deprivation index spent longer in hospital than those with manual occupations living in less deprived areas. Similarly, those with lower education level living in more deprived areas had the greatest risk of hospitalisation. This suggests that hospitalisation is greatest when those with poorer individual socioeconomic factors are combined with residential deprivation. We considered a number of possible explanations for these findings.

### Strengths and limitations of this study

The EPIC-Norfolk cohort is very well characterised. This enabled us to take into account many potentially confounding variables understood to be related to hospital usage and disease. The UK National Health Service is free at the point of use and consequently income is not a major determinant of hospital admissions. Despite this,

social class, education and residential deprivation were all independently related to hospital use. Our study examines hospital activity using a prospective cohort design in a population of community-dwelling participants with clearly defined population denominators. It uses a large cohort of middle-aged and older men and women with 19 years of follow-up time having both area-based census measures and individual social class and education level from questionnaires available.

Townsend Area Deprivation Index is associated with individual sociodemographic factors such as occupational social class and education and other factors including age, sex and BMI. Since all these factors are also related to hospital use, some level of confounding will be present. However, multivariable regression models adjusting for all these variables only modestly attenuated the area deprivation associations. In online supplementary table S2, we stratified by the main confounders and the results remained consistent in the subgroups. The accuracy of the measurement might not be sufficient to ensure adequate adjustment, so we cannot exclude the possibility of residual confounding with known or other unknown factors associated with both Townsend Area Deprivation Index and hospital usage. These unknown factors may either attenuate or strengthen the associations. Interactions between area deprivation and individual sociodemographic factors highlighted stronger associations among more deprived groups.

The use of area-based measurements has some limitations. The factors used in the Townsend score may vary in their ability to assess deprivation according to setting. In urban areas, lower car ownership rates may reflect the availability of other transport options and closer proximity of work places and facilities such as shops. In rural areas, overcrowding may be less common while car ownership may be more of a necessity while simultaneously a drain on resources. The deprivation index is based on data from the UK census that only takes place every 10 years and over the period under examination, areas may change becoming more or less deprived.

Area deprivation was determined by postcode of residence in the year 2000. Study participants who moved house may have been misclassified for some of the period over which hospitalisation was assessed. However, while 22% of the cohort moved house between the years 2000 and 2014, the large majority of participants relocated locally in Norfolk, with others moving elsewhere in England and Wales. Since the Townsend Area Deprivation Index was not measured at enumeration district level in the UK census beyond 1991, no directly comparable measure was available at later time points to examine change. However, a sensitivity analysis of non-movers found very similar results to the main analyses and any misclassification due to moves or changes over time in residential area deprivation scoring and resultant measurement error would only be likely to attenuate associations with the residential area score. HES record were available for participants who relocated within

England and Wales and hence there was virtually no loss to follow-up.

Differential misclassification in hospital use may be explained by early death rates. Study participants living in more deprived areas may have died earlier and not used hospital services for the full period. However, while the death rate was higher among those living in the most deprived areas, 65% of the cohort survived beyond 2018 and models restricted to survivors were more strongly associated with outcome measures than those in the main analysis. Sociodemographic factors may be less relevant for the very seriously ill who require hospital treatment at the end of life.

It may also be possible that individuals did not use NHS facilities but private hospitals differently by socioeconomic status which might explain lower use in the higher sociodemographic groups. However, the use of private hospitals in the Norfolk area over this time period was minimal[27] and hence record linkage of routinely collected hospital episode data gave virtually complete ascertainment. Reverse causation is also possible whereby those in poor health at recruitment may have lower occupational social class increasing the chance of them living in a more deprived area. However, hospitalisation rates were low in the period directly after recruitment.

## Comparison with other studies

Inequality in healthcare favouring the better off has been observed in many countries[10–12] and healthcare insurance and eligibility for government healthcare based on income thresholds may influence the associations observed. NHS healthcare is not constrained by ability to pay and hence we were able to examine the independent association of residential area deprivation—material living conditions, poor quality housing and poor infrastructure—and its association with subsequent hospitalisation.

There is some evidence to suggest that travel time is associated with hospital use,[31 32] but there was no strong association in this study. Study participants were approximately evenly divided into those living in urban and those in rural areas. The moderately deprived (those with Townsend quintile 2–4) were more likely to live in rural areas while the most deprived (Townsend quintile 5) were predominantly urban dwellers. Study participants in Townsend quintiles 1 and 5 were closer by road from their home to the Norfolk and Norwich hospital but the time taken for the journey did not vary greatly. Neither distance from hospital nor urban or rural location explained our findings, since those in the lowest deprivation areas are mainly urban with the shortest travel time to hospital. Studies examining urban/rural populations and car ownership have noted differences in deprivation characteristics.[33 34] However, irrespective of travel distance or time, owning or having access to a car would make a considerable difference in being able to access local facilities. Although there may be more regular public transport services in cities, this will vary and cost and limited travel options may restrict access to hospital and to friends and

relatives, to better quality supermarkets and to parks and recreational facilities.

Most studies examining deprivation in the context of health, disease and mortality either rely on area-based measures collected, for example, from census data[16–20] or from individual level data from questionnaires.[13 14] We had access to both forms of information, having derived individual social class and education level from self-reported questionnaires and area level measures from residential postcode linkage. Hospital-based studies using patients as study participants do not have a reliable population-based denominator and cannot estimate overall risk in the population. Studies often attempt to define a denominator using separate population estimates while not individually linking.[16 35 36] We were able to examine hospital usage over 19 years in a clearly defined community-based population using a prospective cohort design.

Norfolk is an area of generally low deprivation with >80% of the study population living in areas with deprivation levels below the national average. Few participants live in areas of high deprivation such as those found in some larger cities in other parts of the country. Those living in more deprived cities or regions have a socioeconomic gradient in hospital usage more extreme than we were able to observe[37] but while our study does not provide any information on the most extreme forms of deprivation, there was sufficient heterogeneity to observe large differences in hospital use.

Our results provide further evidence adding to the substantial literature linking deprivation to health. Unlike many studies, we used overall measures of hospital activity, including both elective and emergency admissions and found evidence of an independent association of residential area deprivation not accounted for by known individual factors such as social class and education. Our results also demonstrate that the combination of residential area deprivation with lower levels of education or manual social class result in the highest levels of hospitalisation.

The Black report[2 3] concluded that health inequalities were not mainly attributable to failings in the NHS, but rather to many other social inequalities influencing health: income, education, housing, diet, employment and conditions of work. It suggested two mechanisms for how social determinants influence health: cultural/behavioural and materialist/structuralist. Some authors have pointed out that research on the determinants of health are generally focused on the individual but patterns of population health are unclear without examining structural determinants at the societal level.[38] Townsend's residential deprivation index uses aggregate measures of particular characteristics for people living in an area. It has been used mainly as a surrogate for individual measures of deprivation in many studies.[21] We were not able to examine physical features of the environment in this study. Ecological measurements such as the quality of housing, access to recreational facilities, local services provided, community support and levels of crime may affect health and hospital usage. However, we were able to examine both individual and area level deprivation in the same study participants, and the interaction we observed suggests that there is a higher risk of hospitalisation in more deprived areas of residence disproportionately for those with lower individual social class and education. Conversely, individuals with non-manual social class and higher levels of education appear more resilient to hospitalisation irrespective of the level of deprivation of their residence.

## CONCLUSIONS AND POLICY IMPLICATIONS

There is a socioeconomic gradient in hospital usage for factors measured both individually and at area level. Residential area deprivation predicts future hospitalisations, time spent in hospital and number of admissions, independently of individual social class and education level and other behavioural factors. There are significant interactions such that residential area deprivation has greater impact in those with low education level or manual social class. Conversely, higher education level and social class mitigated the association of area deprivation with hospital usage. Effective NHS and government policy should therefore involve addressing deprivation both at the individual and infrastructural levels to identify and target those most at risk within the community. NHS policies focused on reducing health inequalities in the elderly need to work alongside wider government initiatives to improve the quality of housing, transport and infrastructure and access to recreation and green space.

**Acknowledgements** The authors would like to thank the participants, general practitioners and staff of EPIC-Norfolk.

**Contributors** K-TK, NW SH and RL were involved in the conception and design of the study. RL drafted the manuscript, with support from K-TK and PPP. AK contributed to data interpretation. RL was responsible for external data linkage. SH and RL contributed to data collection and acquisition. All authors read and critically revised the manuscript and approved the final manuscript. RL is the guarantor. The corresponding author attests that all listed authors meet authorship criteria and that no others meeting the criteria have been omitted.

**Funding** EPIC-Norfolk infrastructure and core functions are supported by grants from the Medical Research Council (G0401527) and Cancer Research UK (C864/A8257).

**Competing interests** RL, SH, K-TK and NW report grants from MRC and CRUK during the conduct of the study.

**Patient and public involvement statement** The EPIC-Norfolk study have an active Participants Advisory Panel, which meets quarterly to advise on research protocols, suggest ideas and provide feedback on the research including proposed new studies and collaborations. All participants of the EPIC-Norfolk study are informed about the study through regular newsletters as well as public meetings. Information is also disseminated through local community talks in the Norfolk area and science festivals.

**Patient consent for publication** Not required.

**Ethics approval** The work was approval by the Norfolk Research Ethics Committee (98CN01). All participants gave informed signed consent for study participation including access to medical records

**Provenance and peer review** Not commissioned; externally peer reviewed.

**Data availability statement** Data from the EPIC Norfolk cohort are available upon request to the study steering group at https://www.epic-norfolk.org.uk/.

**ORCID iD**
Robert Luben http://orcid.org/0000-0002-5088-6343

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
