## [Reviewer comments · BMJ Open]

ARTICLE DETAILS

TITLE (PROVISIONAL)	Residential area deprivation and risk of subsequent hospital admission in a British population: the EPIC-Norfolk cohort
AUTHORS	Luben, Robert; Hayat, Shabina; Khawaja, Anthony; Wareham, Nicholas; Pharoah, Paul P; Khaw, Kay-Tee

VERSION 1 – REVIEW

REVIEWER	R. David Hayward, PhD Ascension Saint John Hospital Detroit, Michigan USA
REVIEW RETURNED	03-Sep-2019

GENERAL COMMENTS	This paper presents an analysis of prospective data on number and length of hospital admission in relation to socioeconomic variables at the neighborhood and individual levels. This study has a number of strengths including a strong dataset, lengthy follow-up period, and an analytic strategy that anticipates and accounts for important confounding factors. My comments are aimed at addressing some points related to manuscript clarity. 1. The “study design” subheading of the Methods section only covers cohort selection at baseline, making the longitudinal component unclear for another two pages. It would enhance the clarity of the methods to include a brief overview of the various data sources and timing at the beginning of the Methods section.2. It is not clear why the interaction analyses are included in the supplementary material (Table S1) instead of within the main text of the results. Given that the significant interaction effects are referenced along with the main results in both the abstract and the discussion, it seems that these are more than “supplementary” from a conceptual standpoint, but part of the primary analyses. I would suggest considering incorporating these with the main text of the paper.3. The implications of the results in Table 3, disaggregating the components of neighborhood deprivation, are not addressed in detail in the discussion section. As such, the rationale for examining the scale components separately is not clearly articulated, and the contribution of this set of analyses to the results as a whole is ambiguous. These analyses could be explained in more detail to give them further context and a stronger justification. Alternately, these analyses might be a better candidate for being treated as a supplemental analysis than the interaction analyses (given that they do not make a big enough impact on the overall picture to feature in the abstract or discussion).
---

REVIEWER	Tessa Jansen Nivel, Netherlands institute for health services research. The Netherlands
REVIEW RETURNED	06-Sep-2019

GENERAL COMMENTS	Clearly, the authors addressed an important and relevant topic, and they conducted a very valuable study. I agree with the authors that a particular strength of the study is the large cohort that provided a delimited population denominator. Moreover, the follow-up time of 19 years was substantial and enabled robust measurement. However, I have some difficulty with the reporting of the study. First of all, I think the introduction and discussion could largely be improved by more thoroughly positioning both the problem statement (which is currently lacking) as well as the findings in the larger context of (internationally oriented) literature. That would also guide the reporting of the results, which currently appears to me as based on random choices. Furthermore, although the authors report a number of limitations, I do have some questions and thoughts about some of these limitations. Also the reporting of the results should be more balanced, specifically in the discussion section and the abstract. I formulated my comments in more detail below. 1.) In the introduction the overall problem of socioeconomic inequality is stated. Subsequently, the authors merely describe the how of the background of the study, but the way is lacking. There is no problem statement, and no aim as to why this study addresses specifically what problem. Furthermore, the used references are UK oriented, which is too narrow for an international journal. The references to literature are, although relevant, outdated and suggest that no research to socioeconomic inequalities has been conducted in the past two decades. I would suggest to refer to more recent literature, which applies for the discussion as well. 2.) The method section is very detailed, could be more brief (for instance, the measurement of BMI could be assumed as common knowledge). How height and weight were measured is quite irrelevant for this particular study. Referring to a method paper would suffice. Why travel distance/time to hospital were included did not became clear to me. Although reported in table 1, they were not included in further analyses. Why was that? 3.) Since a problem statement is lacking, there is no rationale underlying the presentation of the results. For instance, what question is answered by stratifying the results in table 3 to sex? Whether differences according to sex could be expected should be problematized in the introduction. Similarly, although I think it is valuable to study, no rationale was provided to look into the interaction between area deprivation and individual deprivation indicators. Moreover, I find it problematic to report this finding in the remainder of the results section, and in a supplementary table, whereas this finding was presented as one of the main results in the discussion and abstract. Although the interaction was statistically significant, I doubt the relevance since the OR was very small. Subsequently, figure 1 was introduced without further notice, leaving me puzzled about its meaning. - At page 7, line 52/53 is mentioned that 35.0% of study participants had no admissions over 19 years of the study period.
---

	That does not correspond to table 1, which reports 10.5% having no admissions.  - To get a notion of the relevance of the odds ratios, I would suggest to report the number of participants within each deprivation index quintile that had more than 7 admissions and more than 20 days of hospital stay. 3.) In the discussion there is a lot of repetition of the findings and the limitations. For instance, under the heading 'comparison with other studies', no other studies are cited, but only the study methods are evaluated. The interpretation of the findings in the context of literature is marginal, as are the implications for policy (and research). By better positioning in international literature in the introduction, and addition of a clear problem statement and aim, this issue could better be addressed in the discussion as well. To increase the relevance to a larger public than merely the UK based readers, I would gladly read how the findings of the study translate to other settings and other countries. The absence of implications (for policy, as suggested by the heading, but also for research) is a missed opportunity. Although suggested that the free use of NHS hospital yield equitable healthcare, much more factors are associated with healthcare use. These other possible explanations were not problematized in the introduction nor the discussion. 4.) Limitations of this study that were not mentioned are:  - Use of Townsend index at one moment in time (1991, before recruitment of cohort), whereas index was available for every 10 years. Why not used three index for three moments: 1991, 2001, and 2011, of which the second and third fall within the follow-up period. Although a possible change in deprivation was mentioned in the limitations-section, no explanation is provided for not including measures at multiple times. - The health status/underlying morbidity of individuals was unknown. Typically, low SES individuals suffer from more chronic diseases. - Hospital admission was measured over the whole time period, without controlling for age at admission, whereas deprived individuals contract disease at earlier age. - I would not suggest to conduct additional analysis, but it might be valuable to distinct between elective and emergency admissions in possible future research. Emergency admission are likely to be more socioeconomically patterned than elective admissions. - In the box 'strengths and limitations of this study', the fifth bullet point is obsolete. The authors debut the limitation in the discussion. I would suggest to mention a limitation that actually challenged the interpretation of the results. 5.) Additionally, I would suggest the authors to carefully examine their wording and construction of sentences, as some are unclear. Examples (but not inclusive) are:  - Introduction, page 5, line 32/32: 'When measured (...) to a population': unclear connotation. Please reformulate. - Methods, page 5, line 60: 'Briefly (...) health examination': hard to read, please rephrase. - Methods, page 6, line 10-14: hard to read. Please breakup sentences. - Methods, page 6, line 57/58: 'Participants who'. Should be: who's.
--	---

	- Methods, page 7, line 15: End of sentence: leaving 24977. Please add participants. - At page 10, line 7/8: 'However, (...) admissions': unemployment was not most strongly associated, but the only component that was statistically significant associated. Although it seems semantical, the meaning if different.
--	--

REVIEWER	Polina Putrik Monash University, Australia/Maastricht University, NL I was supported by my junior colleague Rachelle Meisters, Maastricht University who has relevant expertise.
REVIEW RETURNED	12-Sep-2019

GENERAL COMMENTS	This study adds to extensive body of published evidence from EPIC-Norfolk cohort. Authors linked the cohort to hospital admissions and evaluated the relationship between the area deprivation, individual SES and hospital admission over 19 years. This study indirectly contributes to the body of evidence on costs of socio-economic deprivation. Few methodological concerns should be addressed by the authors. Discussion and implications of the findings lack depth and need to be revised. Major 1. INTRODUCTION: The motivation for the study objectives is not entirely clear from the background section. Authors correctly point out that socio-economic gradient in mortality is well explored. The relationship between individual and area level deprivation and health outcomes is also reasonably well-documented over the last decades. As morbidity was not taken into account in the models, hospital use is essentially serving the proxy for health status and confirms that people in deprived areas are less healthy. It would, however, be interesting to explore if hospital use is higher in deprived areas after adjusting for morbidity which would indicate that health care use patterns are different in these areas (due to (in)availability of other provides or other factors such as e.g. loneliness when patients are admitted for 'social reasons'). The reviewer appreciates that authors may not have usable data on health status, however, deeper discussion of what (new) can we expect to learn from this study would be very welcome in the introduction. 2. METHODS: a. What was the motivation to use the chosen cut-off values for outcome measures – 7 admissions, 20 days? Have authors considered models that could deal with multi-category outcome, and if so, what were the reasons for not using them in this case? b. Are single components of Townsend index suitable to be used as predictors? Or is it the combination of the items that defines area as deprived? The paper is quite complex to read and these analyses could be omitted unless authors can justify their value. Notably, this is also not further discussed in the discussion. c. Authors report stratified analyses by a number of factors. The motivation for such analyses could be included in the introduction. Stratified analyses should be considered after testing the significance of the interactions (so part of the table S1 could probably be omitted in case interactions are not significant). The procedures followed should be documented in the methods. d. Type of models used should be also described in methods ('statistical analyses').
--

	e. Could you please clarify how were those who moved house included in the analyses? 3. RESULTS: a. P. 12 Authors state ‘the use of private hospitals over this time period was minimal’. Please refer to source of information for this statement. b. Interestingly, results reveal no real gradient observed according to deprivation quintiles – it is the ‘worst’ 5th quintile that has higher OR for more health utilization (and others are more similar). This deserves discussion and eventually more insights as to which areas are included in this last quintile. 4. DISCUSSION: a. It is remarkable that section ‘comparison with other studies’ does not contain a single reference. This part appears rather superficial and better contextualization of findings is warranted. b. Authors could consider discussing their findings in a view of costs of socio-economic deprivation which is a very relevant topic. c. Reverse causation might be worth discussing. I.e. people are (chronically) ill or in poorer health and have a more/longer hospitalizations, but at the same time, their health may also affect their occupational social class and income levels which in turn may result in a higher chance of living in a highly deprived area at the moment of inclusion to cohort. Moderate/minor comments 5. Part ‘Statistical analyses’ contains a mix of methods and results. Namely, data on missing values belongs to results. 6. One of the references used in the introduction (#10) does not seem to support the statement. 7. In table 1, “time spent in hospital” in which units? (presumably days? please specify) 8. Legend under table 1 is unclear as to what it refers to.
--	---

VERSION 1 – AUTHOR RESPONSE

Reviewer: 1

Reviewer Name: R. David Hayward, PhD

*Institution and Country: Ascension Saint John Hospital
 Detroit, Michigan
 USA*

Please state any competing interests or state ‘None declared’: None declared

Please leave your comments for the authors below

This paper presents an analysis of prospective data on number and length of hospital admission in relation to socioeconomic variables at the neighborhood and individual levels. This study has a number of strengths including a strong dataset, lengthy follow-up period, and an analytic strategy that anticipates and accounts for important confounding factors. My comments are aimed at addressing some points related to manuscript clarity.

Thank you for your comments.

1. The “study design” subheading of the Methods section only covers cohort selection at baseline, making the longitudinal component unclear for another two pages. It would enhance the clarity of the methods to include a brief overview of the various data sources and timing at the beginning of the Methods section.

We have included an overview of the data sources and the timing at the start of the methods section.

2. It is not clear why the interaction analyses are included in the supplementary material (Table S1) instead of within the main text of the results. Given that the significant interaction effects are referenced along with the main results in both the abstract and the discussion, it seems that these are more than “supplementary” from a conceptual standpoint, but part of the primary analyses. I would suggest considering incorporating these with the main text of the paper.

Table S1 (now renamed S2) examines the main associations stratified into subgroups primarily to explore potential confounding examining the consistency of the association in different subgroups as well as examining the possibility of interaction. The interaction between social class and area deprivation is presented as a graph in the main text of the paper rather than in tabular form but is now better described.

3. The implications of the results in Table 3, disaggregating the components of neighborhood deprivation, are not addressed in detail in the discussion section. As such, the rationale for examining the scale components separately is not clearly articulated, and the contribution of this set of analyses to the results as a whole is ambiguous. These analyses could be explained in more detail to give them further context and a stronger justification. Alternately, these analyses might be a better candidate for being treated as a supplemental analysis than the interaction analyses (given that they ~~are~~ do not make a big enough impact on the overall picture to feature in the abstract or discussion).

We agree we have not clearly articulated the rationale for examining Townsend components and they do not have a sufficient importance to warrant their inclusion as a main table. We have now put the original table 3 showing the individual components as a supplementary table (table S1).

Reviewer: 2

Reviewer Name: Tessa Jansen

Institution and Country: Nivel, Netherlands institute for health services research. The Netherlands

Please state any competing interests or state 'None declared': None declared

Please leave your comments for the authors below

Clearly, the authors addressed an important and relevant topic, and they conducted a very valuable study. I agree with the authors that a particular strength of the study is the large cohort that provided a delimited population denominator. Moreover, the follow-up time of 19 years was substantial and enabled robust measurement. However, I have some difficulty with the reporting of the study. First of all, I think the introduction and discussion could largely be improved by more thoroughly positioning both the problem statement (which is currently lacking) as well as the findings in the larger context of (internationally oriented) literature. That would also guide the reporting of the results, which currently appears to me as based on random choices. Furthermore, although the authors report a number of limitations, I do have some questions and thoughts about some of these limitations. Also the reporting of the results should be more balanced, specifically in the discussion section and the abstract. I formulated my comments in more detail below.

Thank you for these comments. I have explained below the changes made to address your individual points or reasons for preferring to maintain the original text.

1.) In the introduction the overall problem of socioeconomic inequality is stated. Subsequently, the authors merely describe the how of the background of the study, but the way is lacking. There is no problem statement, and no aim as to why this study addresses specifically what problem. Furthermore, the used references are UK oriented, which is too narrow for an international journal. The references to literature are, although relevant, outdated and suggest that no research to socioeconomic inequalities has been conducted in the past two decades. I would suggest to refer to more recent literature, which applies for the discussion as well.

We have amended the text to make clearer the problem being addressed by the study. The references used are predominantly British. The context for the study is hospital usage of a British cohort in the UK National Health Service (NHS). While generalisable to some extent, there are also many elements specific to a British context where international literature would not be comparable. In particular the UK NHS provides health services free at the point of delivery so unlike many other countries, ability to pay or having health insurance should not limit access to services. We did not aim to examine international health service utilisation which would have been a different study. While we do reference recent papers, the older literature includes many influential works in the field that remain relevant. However, we have now included some additional references including some that make international comparisons.

2.) The method section is very detailed, could be more brief (for instance, the measurement of BMI could be assumed as common knowledge). How height and weight were measured is quite irrelevant for this particular study. Referring to a method paper would suffice. Why travel distance/time to hospital were included did not become clear to me. Although reported in table 1, they were not included in further analyses. Why was that?

Including a description of how the various covariates were collected or measured is important but may not always be of interest to all readers. Area measures such as distance and travel time to hospital, moved house and urban or rural residence were included since they may have partly explained the associations observed. However, we found no evidence of this as discussed later in the paper

(discussion, page 11, line 38).

3.) Since a problem statement is lacking, there is no rationale underlying the presentation of the results. For instance, what question is answered by stratifying the results in table 3 to sex? Whether differences according to sex could be expected should be problematized in the introduction. Similarly, although I think it is valuable to study, no rationale was provided to look into the interaction between area deprivation and individual deprivation indicators. Moreover, I find it problematic to report this finding in the remainder of the results section, and in a supplementary table, whereas this finding was presented as one of the main results in the discussion and abstract. Although the interaction was statistically significant, I doubt the relevance since the OR was very small. Subsequently, figure 1 was introduced without further notice, leaving me puzzled about its meaning.

- At page 7, line 52/53 is mentioned that 35.0% of study participants had no admissions over 19 years of the study period. That does not correspond to table 1, which reports 10.5% having no admissions.

- To get a notion of the relevance of the odds ratios, I would suggest to report the number of participants within each deprivation index quintile that had more than 7 admissions and more than 20 days of hospital stay.

The purpose of table 3 is to provide additional detail relating to the construction of the Townsend index by disaggregating it and examining the components for men and women separately. It is very common to examine sex-specific associations irrespective of the study aims due to the numerous physiological and social differences between men and women. However, table 3 is not of sufficient importance to warrant its inclusion as a main table since its results are not developed later on and do not affect the study conclusions and hence it has been reclassified as a supplementary table.

The examination of the interaction between area deprivation and individual deprivation indicators is a main aim of the study. The rationale for doing the study is now better explained in the text as mentioned previously. The main findings of the study appear in table 2 and (in graphical form) in figure 1. We agree that as a main finding, the section in results about figure 1 should be earlier and more clearly identified as such and this has now been amended with the main finding stated more clearly.

The figure 35.0% on page 7 was a mistake and is now corrected - thank you for spotting this.

The numbers with more than 7 admissions / 20 hospital days now appear in table 1.

3.) In the discussion there is a lot of repetition of the findings and the limitations. For instance, under the heading 'comparison with other studies', no other studies are cited, but only the study methods are evaluated. The interpretation of the findings in the context of literature is marginal, as are the implications for policy (and research). By better positioning in international literature in the introduction, and addition of a clear problem statement and aim, this issue could better be addressed in the discussion as well. To increase the relevance to a larger public than merely the UK based readers, I would gladly read how the findings of the study translate to other settings and other countries.

The absence of implications (for policy, as suggested by the heading, but also for research) is a missed opportunity. Although suggested that the free use of NHS hospital yield equitable healthcare, much more factors are associated with healthcare use. These other possible explanations were not problematized in the introduction nor the discussion.

We agree that the comparison and citation of other studies in the discussion is inadequate and we have made changes to address this. Some international comparisons are now included. We also agree that the implications for policy are important have now been included. Other factors associated with healthcare (such as recreational facilities, local services, levels of crime etc) are mentioned in the discussion but data on these factors was not available to us.

4.) Limitations of this study that were not mentioned are:

- Use of Townsend index at one moment in time (1991, before recruitment of cohort), whereas index was available for every 10 years. Why not used three index for three moments: 1991, 2001, and 2011, of which the second and third fall within the follow-up period. Although a possible change in deprivation was mentioned in the limitations-section, no explanation is provided for not including measures at multiple times.
 - The health status/underlying morbidity of individuals was unknown. Typically, low SES individuals suffer from more chronic diseases.
 - Hospital admission was measured over the whole time period, without controlling for age at admission, whereas deprived individuals contract disease at earlier age.
 - I would not suggest to conduct additional analysis, but it might be valuable to distinct between elective and emergency admissions in possible future research. Emergency admission are likely to be more socioeconomically patterned than elective admissions.
 - In the box 'strengths and limitations of this study', the fifth bullet point is obsolete. The authors debut the limitation in the discussion. I would suggest to mention a limitation that actually challenged the interpretation of the results.
- Townsend index is only available at "enumeration district" level in the 1991 census. In subsequent censuses it was only measured at (the much larger) "ward" area level which is not comparable. This is now mentioned.
- While underlying morbidity is related to social class, our study examines the subsequent risk of hospitalisation within a community population broadly comparable to the UK population. We use number of hospital admissions and length of stay over a fixed period which differs from survival analysis which uses age at the earliest admission it cannot account for multiple events over a long period.
- Unfortunately, data on emergency admission was not available for this study.

5.) Additionally, I would suggest the authors to carefully examine their wording and construction of sentences, as some are unclear. Examples (but not inclusive) are:

- Introduction, page 5, line 32/32: 'When measured (...) to a population': unclear connotation. Please reformulate.
- Methods, page 5, line 60: 'Briefly (...) health examination': hard to read, please rephrase.
- Methods, page 6, line 10-14: hard to read. Please breakup sentences.
- Methods, page 6, line 57/58: 'Participants who'. Should be: who's.
- Methods, page 7, line 15: End of sentence: leaving 24977. Please add participants.
- At page 10, line 7/8: 'However, (...) admissions': unemployment was not most strongly associated, but the only component that was statistically significant associated. Although it seems semantical, the meaning is different.

The sentences you mention have been adjusted slightly to improve clarity. The grammatical error on page 6 line 57/58 has been corrected. The word participants added to sentence on page 7. The sentence on page 10 line 7/8 has been altered.

Reviewer: 3

Reviewer Name: Polina Putrik

Institution and Country: Monash University, Australia/Maastricht University, NL

I was supported by my junior colleague Rachelle Meisters, Maastricht University who has relevant expertise.

Please state any competing interests or state 'None declared': None declared

Please leave your comments for the authors below

This study adds to extensive body of published evidence from EPIC-Norfolk cohort. Authors linked the cohort to hospital admissions and evaluated the relationship between the area deprivation, individual SES and hospital admission over 19 years. This study indirectly contributes to the body of evidence on costs of socio-economic deprivation. Few methodological concerns should be addressed by the authors. Discussion and implications of the findings lack depth and need to be revised.

Thank you for your helpful comments.

Major

1. INTRODUCTION: The motivation for the study objectives is not entirely clear from the background section. Authors correctly point out that socio-economic gradient in mortality is well explored. The relationship between individual and area level deprivation and health outcomes is also reasonably well-documented over the last decades. As morbidity was not taken into account in the models, hospital use is essentially serving the proxy for health status and confirms that people in deprived areas are less healthy. It would, however, be interesting to explore if hospital use is higher in deprived areas after adjusting for morbidity which would indicate that health care use patterns are different in these areas (due to (in)availability of other provides or other factors such as e.g. loneliness when patients are admitted for 'social reasons'). The reviewer appreciates that authors may not have usable data on health status, however, deeper discussion of what (new) can we expect to learn from this study would be very welcome in the introduction.

We have amended the text to make clearer the problem being addressed by the study. The cohort used is representative in many ways of a middle aged and older UK population and is therefore generalisable to the UK population in general. Hospital usage over time periods differs in many ways to baseline health status. Our aim was to determine whether factors such as material living conditions, poor quality housing and poor infrastructure was associated with subsequent hospitalisation over a long fixed period independently of individual factors. However, rather than examine morbidity within the cohort we wanted to use the cohort as a whole and by extension the UK population.

2. METHODS:

- a. What was the motivation to use the chosen cut-off values for outcome measures – 7 admissions, 20 days? Have authors considered models that could deal with multi-category outcome, and if so, what were the reasons for not using them in this case?
- b. Are single components of Townsend index suitable to be used as predictors? Or is it the combination of the items that defines area as deprived? The paper is quite complex to read and these analyses could be omitted unless authors can justify their value. Notably, this is also not further discussed in the discussion.
- c. Authors report stratified analyses by a number of factors. The motivation for such analyses could be included in the introduction. Stratified analyses should be considered after testing the significance of the interactions (so part of the table S1 could probably be omitted in case interactions are not significant). The procedures followed should be documented in the methods.
- d. Type of models used should be also described in methods ('statistical analyses').
- e. Could you please clarify how were those who moved house included in the analyses?

3. RESULTS:

- a. P. 12 Authors state 'the use of private hospitals over this time period was minimal'. Please refer to source of information for this statement.
- b. Interestingly, results reveal no real gradient observed according to deprivation quintiles – it is the 'worst' 5th quintile that has higher OR for more health utilization (and others are more similar). This deserves discussion and eventually more insights as to which areas are included in this last quintile.

2a) The cut-offs of more than 7 admissions and 20 hospital days are arbitrary but represent approximately the highest 20% of the distribution at 10-year follow-up. The two outcomes are distinct

and complementary but we prefer to use them as constructed to be consistent with previous publications.

2b) The single components of Townsend index are not usually used as individual predictors and it is the combined Townsend score that is most powerful. The purpose of table 3 is to provide additional detail relating to the construction of the Townsend index by disaggregating it and examining the components for men and women separately. However, table 3 is not of sufficient importance to warrant its inclusion as a main table since its results are not developed later on and do not affect the study conclusions and hence it has been reclassified as a supplementary table (table S1).

2c) Table S1 (now table S2) examines the main associations stratified into subgroups primarily to explore potential confounding examining the consistency of the association in different subgroups as well as examining the possibility of interaction. Hence, we would prefer to show associations for all the subgroups and not limit the table to those with an interaction.

2d) The modelling approach - logistic regression is now stated in the methods

2e) Participants who moved house between 2000 and 2014 are included in the main analysis -this point is now stated in the methods.

3a) The statement regarding the use of private hospital is now referenced.

3b) The associations between area deprivation and subsequent hospital usage is most apparent in the fifth quintile in the subset with individual level manual social class and low education levels as shown graphically in figure 1. In the discussion (page13, line 1) we mention that the most deprived (fifth quintile) participants are urban and live closer to hospital.

4. DISCUSSION:

a. It is remarkable that section 'comparison with other studies' does not contain a single reference. This part appears rather superficial and better contextualization of findings is warranted.

b. Authors could consider discussing their findings in a view of costs of socio-economic deprivation which is a very relevant topic.

c. Reverse causation might be worth discussing. I.e. people are (chronically) ill or in poorer health and have a more/longer hospitalizations, but at the same time, their health may also affect their occupational social class and income levels which in turn may result in a higher chance of living in a highly deprived area at the moment of inclusion to cohort.

4a) We agree that the findings need to be put into the context of relevant literature. The discussion has been modified to address this and the introduction now better provides a better framing.

4b) Incorporating costings while a very interesting idea would have the effect of over-extending the paper and we feel it would be better to address this in a separate paper.

4c) This point is now incorporated into the discussion.

Moderate/minor comments

5. Part 'Statistical analyses' contains a mix of methods and results. Namely, data on missing values belongs to results.

6. One of the references used in the introduction (#10) does not seem to support the statement.

7. In table 1, "time spent in hospital" in which units? (presumably days? please specify)

8. Legend under table 1 is unclear as to what it refers to.

The sentence showing missing values has been moved to the results. Reference 10 has been altered. The unit (days) has been added to table 1. The legend under table 1 explains the use of round and square brackets to define intervals such as those appearing as part of the heading just below the text 'Quintile1' , 'Quintile 2' etc.

VERSION 2 – REVIEW

REVIEWER	R. David Hayward Ascension Saint John Hospital Detroit, Michigan USA
REVIEW RETURNED	21-Oct-2019

GENERAL COMMENTS	My comments on the initial draft have been fully addressed.
---

REVIEWER	Polina Putrik, PhD Monash University, Australia / Maastricht University, NL I was supported by my junior colleague Rachelle Meisters, Maastricht University who has relevant expertise.
REVIEW RETURNED	05-Nov-2019

GENERAL COMMENTS	Authors have done a good job addressing by carefully considering the comments. The paper has been substantially improved. A minor comment remaining: 1) It is still not clear what is actually meant by “strict inequalities” and “non-strict inequalities” (Table 1)
---

VERSION 2 – AUTHOR RESPONSE

- 1) It is still not clear what is actually meant by “strict inequalities” and “non-strict inequalities” (Table 1)

Response: "strict inequalities" refers to either "greater than" or "less than" when comparing numbers while "non-strict inequalities" refers to "greater than or equal to" or "less than or equal to". When displaying intervals we used a mathematical notation as a shorthand with mixed square and round brackets. For example $[A,B)$ means "greater than or equal to A and less than B". We have modified the footnote of table 1 to make this clearer.